# INTERPRETATION OF NEURAL NETWORKS IS FRAGILE

## ABSTRACT

In order for machine learning to be deployed and trusted in many applications, it is crucial to be able to reliably explain why the machine learning algorithm makes certain predictions. For example, if an algorithm classifies a given pathology image to be a malignant tumor, then the doctor may need to know which parts of the image led the algorithm to this classification. How to interpret black-box predictors is thus an important and active area of research. A fundamental question is: *how much can we trust the interpretation itself?* In this paper, we show that interpretation of deep learning predictions is extremely fragile in the following sense: two perceptively indistinguishable inputs with the *same* predicted label can be assigned very *different* interpretations. We systematically characterize the fragility of the interpretations generated by several widely-used feature-importance interpretation methods (saliency maps, integrated gradient, and DeepLIFT) on ImageNet and CIFAR-10. Our experiments show that even small random perturbation can change the feature importance and new systematic perturbations can lead to dramatically different interpretations without changing the label. We extend these results to show that interpretations based on exemplars (e.g. influence functions) are similarly fragile. Our analysis of the geometry of the Hessian matrix gives insight on why fragility could be a fundamental challenge to the current interpretation approaches.

## 1 INTRODUCTION

Predictions made by machine learning algorithms play an important role in our everyday lives and can affect decisions in technology, medicine, and even the legal system (Rich, 2015; Obermeyer & Emanuel, 2016). As the algorithms become increasingly complex, explanations for why an algorithm makes certain decisions are ever more crucial. For example, if an AI system predicts a given pathology image to be malignant, then the doctor would want to know what features in the image led the algorithm to this classification. Similarly, if an algorithm predicts an individual to be a credit risk, then the lender (and the borrower) might want to know why. Therefore having interpretations for why certain predictions are made is critical for establishing trust and transparency between the users and the algorithm (Lipton, 2016).

Having an interpretation is not enough, however. The explanation itself must be robust in order to establish human trust. Take the pathology predictor; an interpretation method might suggest that a particular section in an image is important for the malignant classification (e.g. that section could have high scores in saliency map). The clinician might then focus on that section for investigation, treatment or even look for similar features in other patients. It would be highly disconcerting if in an extremely similar image, visually indistinguishable from the original and also classified as malignant, a very different section is interpreted as being salient for the prediction. Thus, even if the predictor is robust (both images are correctly labeled as malignant), that the interpretation is fragile would still be highly problematic in deployment.

**Our contributions.** The fragility of *prediction* in deep neural networks against adversarial attacks is an active area of research (Goodfellow et al., 2014; Kurakin et al., 2016; Papernot et al., 2016; Moosavi-Dezfooli et al., 2016). In that setting, fragility is exhibited when two perceptively indistinguishable images are assigned different labels by the neural network. In this paper, we extend the definition of fragility to neural network interpretation. More precisely, we define the interpretation of neural network to be fragile if perceptively indistinguishable images that have the same prediction label by the neural network are given substantially different interpretations. We systematically

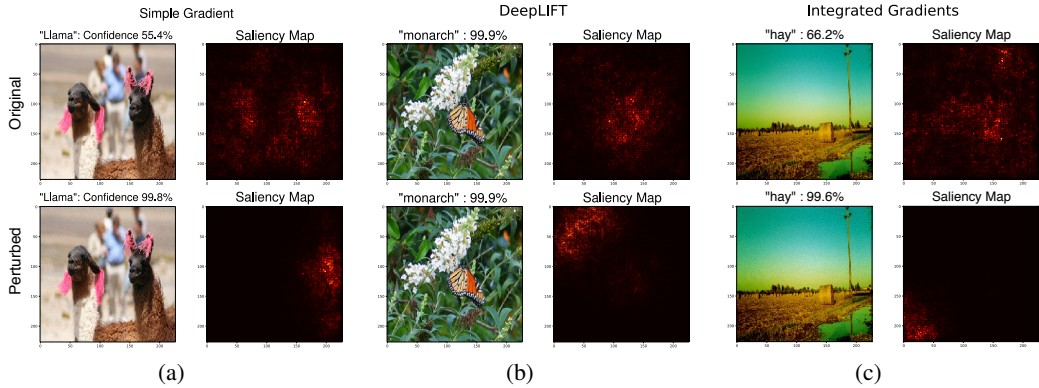

Figure 1: **The fragility of feature-importance maps.** We generate feature-importance scores, also called saliency maps, using three popular interpretation methods: simple gradient (a), DeepLIFT (b) and integrated gradient (c). The **top row** shows the the original images and their saliency maps and the **bottom row** shows the perturbed images (using the center attack with $\epsilon = 8$, as described in Section 3) and the corresponding saliency maps. In all three images, the predicted label has not changed due to perturbation; in fact the network's (SqueezeNet) confidence in the prediction has actually increased. However, the saliency maps of the perturbed images are meaningless.

investigate two classes of interpretation methods: methods that assign importance scores to each feature (this includes simple gradient (Simonyan et al., 2013), DeepLift (Shrikumar et al., 2017), and integrated gradient (Sundararajan et al., 2017)), as well as a method that assigns importances to each training example: influence functions (Koh & Liang, 2017). For both classes of interpretations, we show that targeted perturbations can lead to dramatically different interpretations (Fig. 1).

Our findings highlight the fragility of interpretations of neural networks, which has not been carefully considered in literature. Fragility directly limits how much we can trust and learn from the interpretations. It also raises a significant new security concern. Especially in medical or economic applications, users often take the interpretation of a prediction as containing causal insight (*"this image is a malignant tumor likely because of the section with a high saliency score"*). An adversary could minutely manipulate the input to draw attention away from relevant features or onto his/her desired features. Such attacks might be especially hard to detect as the actual labels have not changed.

While we focus on image data here because most of the interpretation methods have been motivated by images, the fragility of neural network interpretation could be a much broader problem. Fig. 2 illustrates the intuition that when the decision boundary in the input feature space is complex, as is the case with deep nets, a small perturbation in the input can push the example into a region with very different loss contours. Because the feature importance is closely related to the gradient which is perpendicular to the loss contours, the importance scores can also be dramatically different. We provide additional analysis of this in Section 5.

## 2 INTERPRETATION METHODS FOR NEURAL NETWORK PREDICTIONS

### 2.1 FEATURE-IMPORTANCE INTERPRETATION

This first class of methods explains predictions in terms of the relative importance of features in a test input sample. Given the sample $x_t \in \mathbb{R}^d$ and the network's prediction $l$, we define the score of the predicted class $S_l(x_t)$ to be the value of the $l$-th output neuron right before the softmax operation. We take $l$ to be the class with the max score; i.e. the predicted class. Feature-importance methods seek to find the dimensions of input data point that most strongly affect the score, and in doing so, these methods assign an absolute saliency score to each input feature. Here we normalize the scores for each image by the sum of the saliency scores across the features. This ensures that any perturbations that we design change not the absolute feature saliencies (which may still preserve

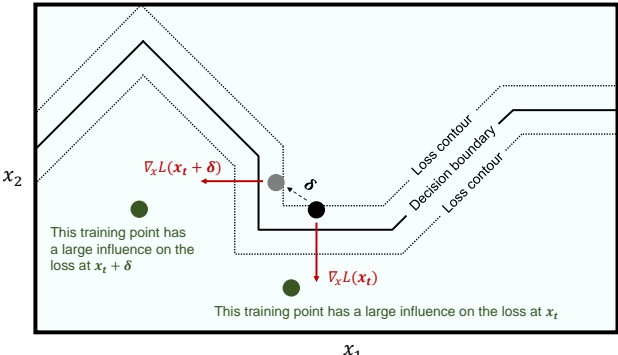

Figure 2: **Intuition for why interpretation is fragile.** Consider a test example $x_t \in \mathbb{R}^2$ (black dot) that is slightly perturbed to a new position $x_t + \delta$ in input space (gray dot). The contours and decision boundary corresponding to a loss function ($L$) for a two-class classification task are also shown, allowing one to see the direction of the gradient of the loss with respect to the input space. Neural networks with many parameters have decision boundaries that are roughly piecewise linear with many transitions. We illustrate that points near the transitions are especially fragile to interpretability-based analysis. A small perturbation to the input changes the direction of $\nabla_x L$ from being in the direction of $x_1$ to being in the direction of $x_2$, directly affecting feature-importance analyses. Similarly, a small perturbation to the test image changes which training image, when up-weighted, has the largest influence on $L$, directly affecting exemplar-based analysis.

the ranking of different features), but their relative values. We summarize three different methods to calculate the normalized saliency score, denoted by $R(x_t)$.

**Simple gradient method**   Introduced in Baehrens et al. (2010) and applied to deep neural networks in Simonyan et al. (2013), the simple gradient method applies a local linear approximation of the model to detect the sensitivity of the score to perturbing each of the input dimensions. Given input $x_t \in \mathbb{R}^d$, the score is defined as:

$$R(x_t)_j = |\nabla_x S_l(x_t)_j| / \sum_{i=1}^{d} |\nabla_x S_l(x_t)_i|. \tag{1}$$

**Integrated gradients**   A significant drawback of the simple gradient method is the saturation problem discussed by Shrikumar et al. (2017); Sundararajan et al. (2017). Consequently, Sundararajan et al. (2017) introduced the integrated gradients method where the gradients of the score with respect to $M$ scaled versions of the input are summed and then multiplied by the input. Letting $x^0$ be the reference point and $\Delta x_t = x_t - x^0$, the feature importance vector is calculated by:

$$R(x_t) = \left| \frac{\Delta x_t}{M} \sum_{k=1}^{M} \nabla_x S_l \left( \frac{k}{M} \Delta x_t + x^0 \right) \right|, \tag{2}$$

which is then normalized for our analysis. Here the absolute value is taken for each dimension.

**DeepLIFT**   DeepLIFT is an improved version of layer-wise relevance propagation (LRP) method (Bach et al., 2015). LRP methods decompose the score $S_l(x_t)$ backwards through the neural network. In each step, the score from the last layer is propagated to the previous layer, with the score being divided proportionally to magnitude of the activations of the neurons in the previous layer. The scores are propagated to the input layer, and the result is a relevance score assigned to each of the input dimensions. DeepLIFT (Shrikumar et al., 2017) defines a reference point in the input space and propagates relevance scores proportionally to the changes in the neuronal activations from the reference. We use DeepLIFT with the Rescale rule; see Shrikumar et al. (2017) for details.

### 2.2   EXEMPLAR-BASED METHODS: INFLUENCE FUNCTIONS

A complementary approach to interpreting the results of a neural network is to explain the prediction of the network in terms of its *training examples*, $\{(x_i, y_i)\}$. Specifically, we can ask: which training

examples, if up-weighted or down-weighted during training time, would have the biggest effect on the loss of the test example $(\boldsymbol{x_t}, y_t)$? Koh & Liang (2017) proposed a method to calculate this value, called the influence, defined by the following equation:

$$I(z_i, z_t) = -\nabla_\theta L(z_t, \hat{\theta})^\top H_{\hat{\theta}}^{-1} \nabla_\theta L(z_i, \hat{\theta}), \tag{3}$$

where $z_i \overset{\text{def}}{=} (\boldsymbol{x}_i, y_i)$ and $z_t$ is defined analogously. $L(z, \hat{\theta})$ is the loss of the network with parameters set to $\hat{\theta}$ for the (training or test) data point $z$. $H_{\hat{\theta}} \overset{\text{def}}{=} \frac{1}{n} \sum_{i=1}^n \nabla_\theta^2 L(z_i, \hat{\theta})$ is the empirical Hessian of the network calculated over the training examples. The training examples with the highest influence are understood as explaining *why* a network made a particular prediction for a test example.

### 2.3 METRICS FOR INTERPRETATION SIMILARITY

We consider two natural metrics for quantifying the similarity between interpretations for two different images. As shown in Fig. 3, these metrics can be used to evaluate the effectiveness of a targeted attack on interpretability.

- **Spearman's rank order correlation**: Because interpretation methods rank all of the features or training examples in order of importance, it is natural to use the rank correlation (Spearman, 1904) to compare the similarity between interpretations.

- **Top-k intersection**: In many settings, only the most important features or interpretations are of interest. In these settings, we can compute the size of the intersection of the $k$ most important features before and after perturbation.

## 3 RANDOM AND SYSTEMATIC PERTURBATIONS

**Problem statement**  For a given fixed neural network $\mathcal{N}$ and input data point $\boldsymbol{x}_t$, the feature importance and influence function methods that we have described produce an interpretation $\boldsymbol{I}(\boldsymbol{x}_t; \mathcal{N})$. For feature importance, $\boldsymbol{I}(\boldsymbol{x}_t; \mathcal{N})$ is a vector of feature scores; for influence function $\boldsymbol{I}(\boldsymbol{x}_t; \mathcal{N})$ is a vector of scores for training examples. We would like to devise efficient perturbations to change the interpretability of a test image. Yet, the perturbations should be visually imperceptible and should not change the label of the prediction. Formally, we define the problem as:

$$\arg\max_{\boldsymbol{\delta}} \mathcal{D}\left(\boldsymbol{I}(\boldsymbol{x}_t; \mathcal{N}), \boldsymbol{I}(\boldsymbol{x}_t + \boldsymbol{\delta}; \mathcal{N})\right)$$
$$\text{subject to: } ||\boldsymbol{\delta}||_\infty \le \epsilon, \text{Prediction}(\boldsymbol{x}_t + \boldsymbol{\delta}; \mathcal{N}) = \text{Prediction}(\boldsymbol{x}_t; \mathcal{N})$$

where $\mathcal{D}(\cdot)$ measures the change in interpretation (e.g. how many of the top-$k$ pixels are no longer the top-$k$ pixels of the saliency map after the perturbation) and $\epsilon > 0$ constrains the norm of the perturbation. In this paper, we carry out three kinds of input perturbations.

**Random sign perturbation**  As a baseline, we generate random perturbations in which each pixel is randomly perturbed by $\pm\epsilon$. This is used to measure robustness against untargeted perturbations.

**Iterative attacks against feature-importance methods**  In Algorithm 1 we define two adversarial attacks against feature-importance methods, each of which consists of taking a series of steps in the direction that maximizes a differentiable dissimilarity function between the original and perturbed interpretation. (1) The **top-k** attack seeks to perturb the saliency map by decreasing the relative importance of the $k$ most important features of the original image. (2) When the input data are images, the center of mass of the saliency map often captures the user's attention. The **mass-center** attack is designed to result in the maximum spatial displacement of the center of mass of the saliency scores. Both of these attacks can be applied to any of the three feature-importance methods.

**Gradient sign attack against influence functions**  We can obtain effective adversarial images for influence functions without resorting to interative procedures. We linearize (3) around the values of the current inputs and parameters. If we further constrain the $L_\infty$ norm of the perturbation to $\epsilon$, we obtain an optimal single-step perturbation:

$$\boldsymbol{\delta} = \epsilon\text{sign}(\nabla_{\boldsymbol{x}_t} I(z_i, z_t)) = -\epsilon\text{sign}(\nabla_{\boldsymbol{x}_t} \nabla_\theta L(z_t, \hat{\theta})^\top \underbrace{H_{\hat{\theta}}^{-1} \nabla_\theta L(z_i, \hat{\theta})}_{\text{independent of } \boldsymbol{x}_t}). \tag{4}$$

---

**Algorithm 1** Iterative Feature-Importance Attacks

---

**Input:** test image $\boldsymbol{x}_t$, maximum norm of perturbation $\epsilon$, normalized feature importance function $\boldsymbol{R}(\cdot)$, number of iterations $P$, step size $\alpha$

Define a dissimilarity function $D$ to measure the change between interpretations of two images:

$$D(\boldsymbol{x}_t, \boldsymbol{x}) = \begin{cases} -\sum_{i \in B} \boldsymbol{R}(\boldsymbol{x})_i & \text{for } \textbf{top-k} \text{ attack} \\ \\ ||\boldsymbol{C}(\boldsymbol{x}) - \boldsymbol{C}(\boldsymbol{x}_t)||_2 & \text{for } \textbf{mass-center} \text{ attack,} \end{cases}$$

where $B$ is the set of the $k$ largest dimensions[a] of $\boldsymbol{R}(\boldsymbol{x}_t)$, and $\boldsymbol{C}(\cdot)$ is the center of saliency mass[b].

Initialize $\boldsymbol{x}^0 = \boldsymbol{x}_t$
**for** $p \in \{1, \ldots, P\}$ **do**

    Perturb the test image in the direction of signed gradient[c] of the dissimilarity function:

$$\boldsymbol{x}^p = \boldsymbol{x}^{p-1} + \alpha \cdot \text{sign}(\nabla_{\boldsymbol{x}} D(\boldsymbol{x}_t, \boldsymbol{x}^{p-1}))$$

    If needed, clip the perturbed input to satisfy the norm constraint: $||\boldsymbol{x}^p - \boldsymbol{x}_t||_\infty \leq \epsilon$
**end for**
Among $\{\boldsymbol{x}^1, \ldots, \boldsymbol{x}^P\}$, return the element with the largest value for the dissimilarity function and the same prediction as the original test image.

---

[a]The goal is to damp the saliency scores of the $k$ features originally identified as the most important.
[b]The center of mass is defined for a $W \times H$ image as:

$$\boldsymbol{C}(\boldsymbol{x}) \overset{\text{def}}{=} \sum_{i \in \{1, \ldots, W\}} \sum_{j \in \{1, \ldots, H\}} \boldsymbol{R}(\boldsymbol{x})_{i,j}[i, j]^T$$

[c]In some networks, such as those with ReLUs, this gradient is always 0. To attack interpretability in such networks, we replace the ReLU activations with their smooth approximation (softplus) when calculating the gradient and generate the perturbed image using this approximation. The perturbed images that result are effective adversarial attacks against the original ReLU network, as discussed in Section 4.

---

The attack we use consists of applying the negative of the perturbation in (4) to decrease the influence of the 3 most influential training images of the original test image[1]. Of course, this affects the influence of all of the other training images as well.

We follow the same setup for computing the influence function as was done by the authors of Koh & Liang (2017). Because the influence is only calculated with respect to the parameters that change during training, we calculate the gradients only with respect to parameters in the final layer of our network (InceptionNet, see Section 4). This makes it feasible for us to compute (4) exactly, but it gives us the perturbation of the input *into the final layer*, not the first layer. So, we use standard back-propagation to calculate the corresponding gradient for the input test image. We then take the sign of this gradient as the perturbation and clip the image to produce the adversarial test image.

## 4 EXPERIMENTS & RESULTS

**Data sets and models** To evaluate the robustness of feature-importance methods, we used two image classification data sets: ILSVRC2012 (ImageNet classification challenge data set) (Russakovsky et al., 2015) and CIFAR-10 (Krizhevsky, 2009). For the ImageNet classification data set, we used a pre-trained SqueezeNet[2] model introduced by Iandola et al. (2016). For the CIFAR-10 data set we trained our own convolutional network, whose architecture is presented in Appendix A.

---

[1]In other words, we generate the perturbation given by: $-\epsilon \text{sign}(\sum_{i=1}^{3} \nabla_{\boldsymbol{x}_t} \nabla_\theta L(z_t, \hat{\theta})^\top H_{\hat{\theta}}^{-1} \nabla_\theta L(z_{(i)}, \hat{\theta}))$, where $z_{(i)}$ is the $i^{\text{th}}$ most influential training image of the original test image.
[2]https://github.com/rcmalli/keras-squeezenet

For both data sets, the results are examined using simple gradient, integrated gradients, and DeepLIFT feature importance methods. For DeepLIFT, we used the pixel-wise and the channel-wise mean images as the CIFAR-10 and ImageNet reference points respectively. For the integrated gradients method, the same references were used with parameter M=100. We ran all iterative attack algorithms for $P = 300$ iterations with step size $\alpha = 0.5$.

To evaluate the robustness of influence functions, we followed a similar experimental setup to that of the original authors: we trained an InceptionNet v3 with all but the last layer frozen (the weights were pre-trained on ImageNet and obtained from Keras[3]). The last layer was trained on a binary flower classification task (**roses** vs. **sunflowers**), using a data set consisting of 1,000 training images[4]. This data set was chosen because it consisted of images that the network had not seen during pre-training on ImageNet. The network achieved a validation accuracy of 97.5% on this task.

**Results for feature-importance methods**   From the ImageNet test set, 512 correctly-classified images were randomly sampled for evaluation purposes. Examples of the mass-center attack against three feature importance methods were presented in Fig. 1. Further representative examples of different attacks on additional images are found in Appendix B.

Fig. 3 is a representative example to illustrate how the decrease in rank order correlation and top-1000 intersection relate to visual changes in the saliency maps. More examples of different methods are pictured in Appendix C.

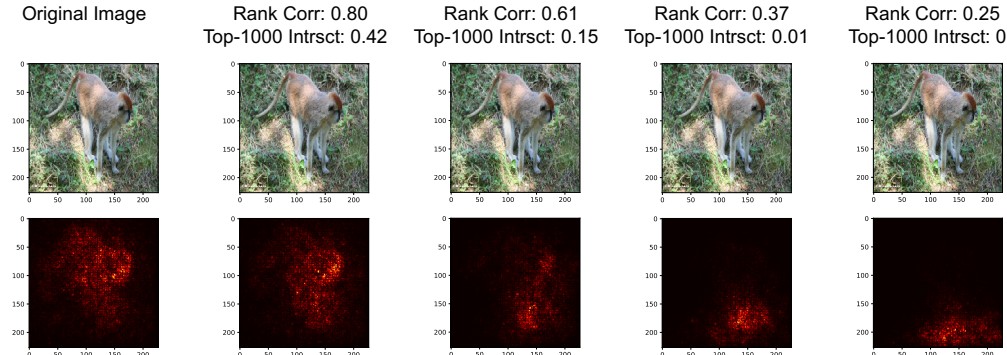

Figure 3: **Evaluation metrics vs subjective change** We generate snapshots of the perturbed image and its simple gradient saliency maps along with iterations of mass-center attack to visualize the gradual change in saliency map with its corresponding the rank-correlation and top-1000 intersection metrics.

In Fig. 4, we present results aggregated over all 512 images. We compare different attack methods using top-1000 intersection and rank correlation methods. In all the images, the attacks does not change the original predicted label of the image. Random sign perturbation already causes significant changes in both top-1000 intersection and rank order correlation. For example, with $L_\infty = 8$, on average, there is less than 30% overlap in the top 1000 most salient pixels between the original and the randomly perturbed images across all three of interpretation methods. This suggests that the saliency of individual or small groups of pixels can be extremely fragile to the input and should be interpreted with caution. With targeted perturbations, we observe more dramatic fragility. Even with a perturbation of $L_\infty = 2$, the interpretations change significantly. Both iterative attack algorithms have similar effects on feature importance of test images when measured on the basis of rank correlation or top-1000 intersection. In Appendix D, we show an additional metric: the displacement of the center of mass between the original and perturbed saliency maps. Empirically, we find this metric to correspond most strongly with intuitive perceptions of the similarity between two saliency maps. Not surprisingly, we found that the center attack method was more effective than the top-$k$ attack at moving the center of mass of the saliency maps. Comparing the fragility of neural network interpretation among the three different methods, we found that the integrated gradients method was

---

[3]https://keras.io/applications/

[4]adapted from: https://www.tensorflow.org/tutorials/image_retraining

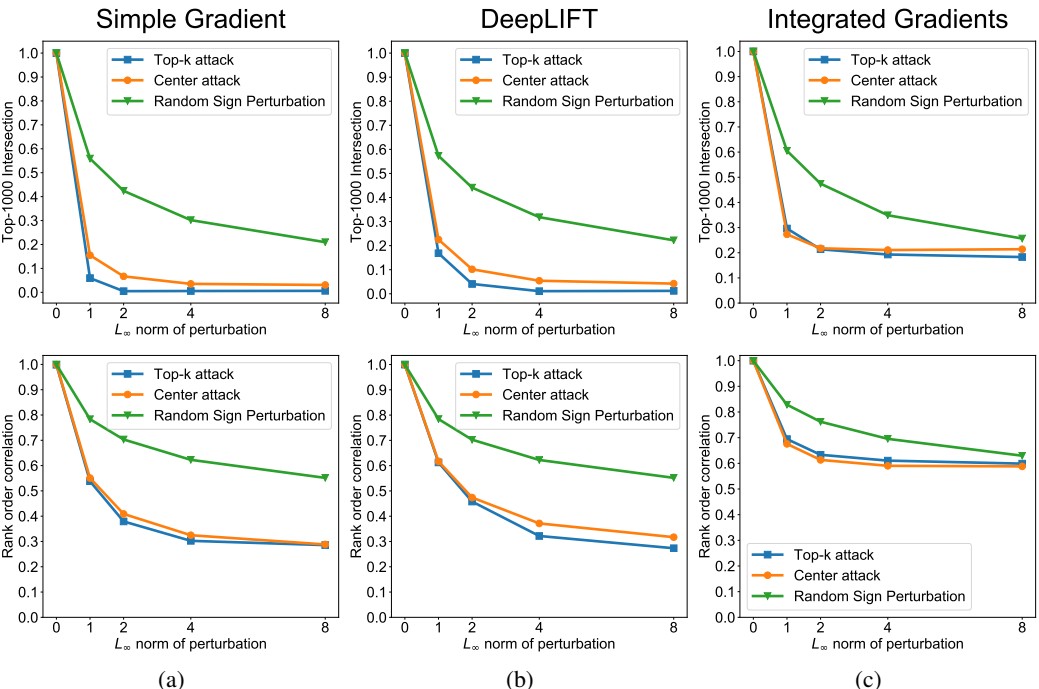

Figure 4: **Comparison of adversarial attack algorithms on feature-importance methods.** Across 512 correctly-classified ImageNet images, we find that the top-$k$ and center attacks perform similarly in top-1000 intersection and rank correlation measures, and are far more effective than the random sign perturbation at demonstrating the fragility of interpretability, as characterized through top-1000 intersection **(top)** as well as rank order correlation **(bottom)**. This is true for (a) the simple gradient method, (b) DeepLift, and (c) the integrated gradients method.

the most robust to both random and adversarial attacks. Similar results for CIFAR-10 can be found in Appendix D.

**Results for influence functions**    We evaluate the robustness of influence functions on a test data set consisting of 200 images of roses and sunflowers. Fig. 5 shows a representative test image to which we have applied the gradient sign attack. Although the prediction of the image does not change, the most influential training examples selected according to (3), as explanation for the prediction, change entirely from images of sunflowers and yellow petals that resemble the input image to those of red and pink roses that do not. Additional examples can be found in Appendix E.

In Fig. 6, we compare the random perturbations and gradient sign attacks across all of the test images. We find that the gradient sign-based attacks are significantly more effective at decreasing the rank correlation of the influence of the training images, as well as distorting the top-5 influential images. For example, on average, with a targeted perturbation of magnitude $\epsilon = 8$, only 2 of the top 5 most influential training images remain as the top 5 most influential images after the visually imperceptible perturbation. The influences of the training images before and after an adversarial attack are essentially uncorrelated. However, we find that even random attacks can have a non-negligible effect on influence functions, on average reducing the rank correlation to 0.8 ($\epsilon \approx 10$).

## 5  HESSIAN ANALYSIS

In this section, we try to understand the source of interpretation fragility. The question is whether fragility a consequence of the complex non-linearities of a deep network or a characteristic present even in high-dimensional linear models, as is the case for adversarial examples for prediction (Goodfellow et al., 2014). To gain more insight into the fragility of gradient based interpretations, let $S(\boldsymbol{x}; \boldsymbol{W})$ denote the score function of interest; $\boldsymbol{x} \in \mathbb{R}^d$ is an input vector and $\boldsymbol{W}$ is the weights of the neural network, which is fixed since the network has finished training. We are interested in the

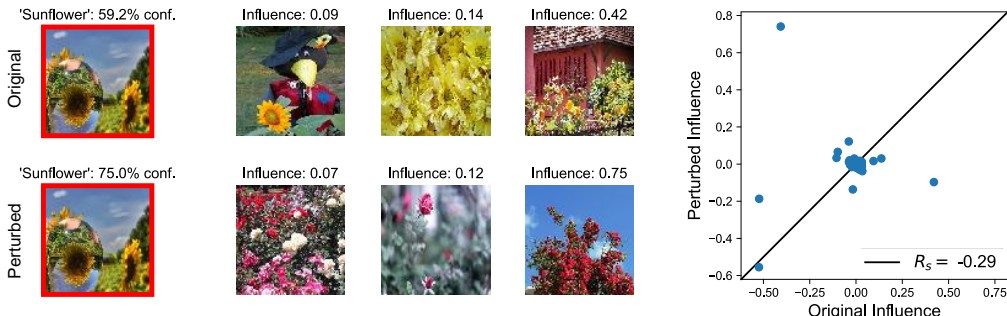

Figure 5: **Gradient sign attack on influence functions.** An imperceptible perturbation to a test image can significantly affect exemplar-based interpretability. The original test image is that of a sunflower that is classified correctly in a rose vs. sunflower classification task. The top 3 training images identified by influence functions are shown in the top row. Using the gradient sign attack, we perturb the test image (with $\epsilon = 8$) to produce the leftmost image in the second row. Although the image is even more confidently predicted as a sunflower, influence functions suggest very different training images by means of explanation: instead of the sunflowers and yellow petals that resemble the input image, the most influential images are pink/red roses. The plot on the right shows the influence of each training image before and after perturbation. The 3 most influential images (targeted by the attack) have decreased in influence, but the influences of other images have also changed.

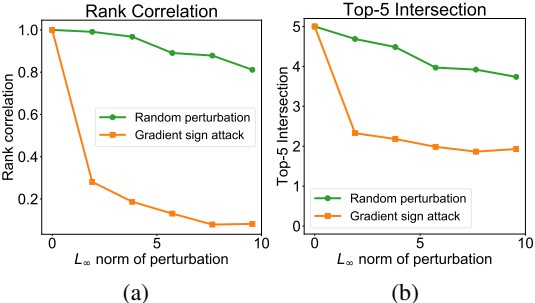

Figure 6: **Comparison of random and targeted perturbations on influence functions.** Here, we show the averaged results of applying random (green) and gradient sign-based (orange) perturbations to 200 test images on the flower classification task. While random attacks affect interpretability, the effect is small and generally doesn't affect the most influential images. On the other hard, a targeted attack can significantly affect (a) the rank correlation and (b) even change the make-up of the 5 most influential images. Even at the maximal level of noise, the changes to the perturbed images were visually imperceptible, and prediction confidence was not significantly changed (the mean change was $< 1\%$ for random attacks and $< 5\%$ for targeted attacks at the highest level of noise).

Hessian $H$ whose entries are $H_{i,j} = \frac{\partial S}{\partial x_i \partial x_j}$. The reason is that the first order approximation of gradient for some input perturbation direction $\boldsymbol{\delta} \in \mathbb{R}^d$ is: $\nabla_{\boldsymbol{x}} S(\boldsymbol{x} + \boldsymbol{\delta}) - \nabla_{\boldsymbol{x}} S(\boldsymbol{x}) \approx H\boldsymbol{\delta}$.

First, consider a linear model whose score for an input $\boldsymbol{x}$ is $S = \boldsymbol{w}^\top \boldsymbol{x}$. Here, $\nabla_{\boldsymbol{x}} S = \boldsymbol{w}$ and $\nabla_{\boldsymbol{x}}^2 S = 0$; the feature-importance vector $\boldsymbol{w}$ is robust, because it is completely independent of $\boldsymbol{x}$. Thus, some non-linearity is required for interpretation fragility. A simple network that is susceptible to adversarial attacks on interpretations consists of a set of weights connecting the input to a single neuron followed by a non-linearity (e.g. softmax): $S = g(\boldsymbol{w}^\top \boldsymbol{x})$.

We can calculate the change in saliency map due to a small perturbation in $\boldsymbol{x} \rightarrow \boldsymbol{x} + \boldsymbol{\delta}$. The first-order approximation for the change in saliency map will be equal to : $H \cdot \boldsymbol{\delta} = \nabla_{\boldsymbol{x}}^2 S \cdot \boldsymbol{\delta}$. In particular, the saliency of the $i^{\text{th}}$ feature changes by $(\nabla_{\boldsymbol{x}}^2 S \cdot \boldsymbol{\delta})_i$ and furthermore, the relative change

is $(\nabla_{\boldsymbol{x}}^2 S \cdot \boldsymbol{\delta})_i / (\nabla_{\boldsymbol{x}} S)_i$. For the simple network, this relative change is:

$$\frac{(\boldsymbol{ww}^\top \boldsymbol{\delta} g''(\boldsymbol{w}^\top \boldsymbol{x}))_i}{(\boldsymbol{w} g'(\boldsymbol{w}^\top \boldsymbol{x}))_i} = \frac{w_i \boldsymbol{w}^\top \boldsymbol{\delta} g''(\boldsymbol{w}^\top \boldsymbol{x})}{w_i g'(\boldsymbol{w}^\top \boldsymbol{x})} = \frac{\boldsymbol{w}^\top \boldsymbol{\delta} g''(\boldsymbol{w}^\top \boldsymbol{x})}{g'(\boldsymbol{w}^\top \boldsymbol{x})}, \tag{5}$$

where we have used $g'(\cdot)$ and $g''(\cdot)$ to refer to the first and second derivatives of $g(\cdot)$. Note that $g'(\boldsymbol{w}^\top \boldsymbol{x})$ and $g''(\boldsymbol{w}^\top \boldsymbol{x})$ do not scale with the dimensionality of $\boldsymbol{x}$ because in general, independent from the dimensionality, $\boldsymbol{x}$ and $\boldsymbol{w}$ are $\ell_2$-normalized or have fixed $\ell_2$-norm due to data preprocessing and weight decay regularization. However, if we choose $\boldsymbol{\delta} = \epsilon \text{sign}(\boldsymbol{w})$, then the relative change in the saliency grows with the dimension, since it is proportional to the $\ell_1$-norm of $\boldsymbol{w}$. When the input is high-dimensional—which is the case with images—the relative effect of the perturbation can be substantial. Note also that this perturbation is exactly the sign of the first right singular vector of the Hessian $\nabla_{\boldsymbol{x}}^2 S$, which is appropriate since that is the vector that has the maximum effect on the gradient of $S$. A similar analysis can be carried out for influence functions (see Appendix F).

For this simple network, the direction of adversarial attack on interpretability, $\text{sign}(\boldsymbol{w})$ is the same as the adversarial attack on prediction. This means that we cannot perturb interpretability independently of prediction. For more complex networks, this is not the case and in Appendix G we show this analytically for a simple case of a two-layer network. As an empirical test, in Fig. 7(a), we plot the distribution of the angle between $\nabla_{\boldsymbol{x}} S$ and $\boldsymbol{v}_1$ (the first right singular vector of $H$ which is the most fragile direction of feature importance) for 1000 CIFAR10 images (Details of the network in Appendix A). In Fig. 7(b), we plot the equivalent distribution for influence functions, computed across all 200 test images. The result confirms that the steepest direction of change in interpretation and prediction are generally orthogonal, justifying how the perturbations can change the interpretation without changing the prediction.

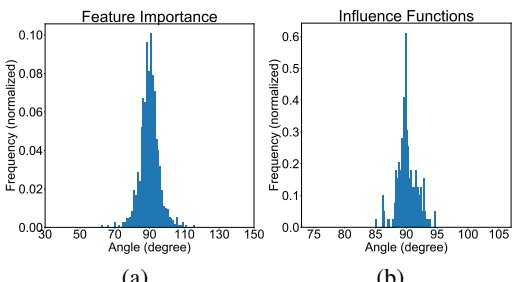

(a)        (b)

Figure 7: **Orthogonality of Prediction and Interpretation Fragile Directions** (a) The histogram of the angle between the steepest direction of change in feature importance and the steepest score change direction. (b) The distribution of the angle between the gradient of the loss function and the steepest direction of change of influence of the most influential image.

## 6 DISCUSSION

**Related works** To the best of our knowledge, the notion of adversarial examples has not previously been studied in the context of interpretation of neural networks. Adversarial attacks to the input that changes the *prediction* of a network have been actively studied. Szegedy et al. (2013) demonstrated that it is relatively easy to fool neural networks into making very different predictions for test images that are visually very similar to each other. Goodfellow et al. (2014) introduced the Fast Gradient Sign Method (FGSM) as a one-step prediction attack. This was followed by more effective iterative attacks (Kurakin et al., 2016) seeking to change the prediction of network by a small perturbation. Different metrics for quantifying the size of the perturbation have been used. Moosavi-Dezfooli et al. (2016); Szegedy et al. (2013) used $\ell_2$; Papernot et al. (2016) considered the number of perturbed pixels ($\ell_0$); and Goodfellow et al. (2014) suggest using $\ell_\infty$, because this tightly controls how much individual feature can change. We followed the popular practice and evaluate with $\ell_\infty$.

Interpretation of neural network predictions is also an active research area. Post-hoc interpretability (Lipton, 2016) is one family of methods that seek to "explain" the prediction without talking about the details of black-box model's hidden mechanisms. These included tools to explain predictions by networks in terms of the features of the test example (Simonyan et al., 2013; Shrikumar et al., 2017; Sundararajan et al., 2017; Zhou et al., 2016), as well as in terms of contribution of training examples to the prediction at test time (Koh & Liang, 2017). These interpretations have gained increasing popularity, as they confer a degree of insight to human users of what the neural network might be doing (Lipton, 2016).

**Conclusion** This paper demonstrates that interpretation of neural networks can be fragile in the specific sense that two similar inputs with the same predicted label can be given very different interpretations. We develop new perturbations to illustrate this fragility and propose evaluation metrics as well as insights on why fragility occurs. Fragility of neural network interpretation is orthogonal to fragility of the prediction—we demonstrate how perturbations can substantially change the interpretation without changing the predicted label. The two types of fragility do arise from similar factors, as we discuss in Section 5. Our focus is on the interpretation method, rather than on the original network, and as such we do not explore how interpretable is the original predictor. There is a separately line of research that tries to design simpler and more interpretable prediction models (Ba & Caruana, 2014).

Our main message is that robustness of the interpretation of a prediction is an important and challenging problem, especially as in many applications (e.g. many biomedical and social settings) users are as interested in the interpretation as in the prediction itself. Our results raise concerns on how interpretations of neural networks are sensitive to noise and can be manipulated. Especially in settings where the importance of individual or a small subset of features are interpreted, we show that these importance scores can be sensitive to even random perturbation. More dramatic manipulations of interpretations can be achieved with our targeted perturbations, which raise security concerns. We do not suggest that interpretations are meaningless, just as adversarial attacks on predictions do not imply that neural networks are useless. Interpretation methods do need to be used and evaluated with caution while applied to neural networks, as they can be fooled into identifying features that would not be considered salient by human perception.

Our results demonstrate that the *interpretations* (e.g. saliency maps) are vulnerable to perturbations, but this does not imply that the *interpretation methods* are broken by the perturbations. This is a subtle but important distinction. Methods such as saliency measure the infinitesimal sensitivity of the neural network at a particular input $x$. After a perturbation, the input has changed to $\tilde{x} = x + \delta$, and the salency now measures the sensitivity at the perturbed input. The saliency *correctly* captures the infinitesimal sensitivity at the two inputs; it's doing what it is supposed to do. The fact that the two resulting saliency maps are very different is fundamentally due to the network itself being fragile to such perturbations, as we illustrate with Fig. 2.

While we focus on image data (ImageNet and CIFAR-10), because these are the standard benchmarks for popular interpretation tools, this fragility issue can be wide-spread in biomedical, economic and other settings where neural networks are increasingly used. Understanding interpretation fragility in these applications and develop more robust methods are important agendas of research.

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

APPENDICES

## A   DESCRIPTION OF THE CIFAR-10 CLASSIFICATION NETWORK

We trained the following structure using ADAM optimizer  (Kingma & Ba, 2014) with default parameters.  The resulting test accuracy using ReLU activation was 73%.  For the experiment in Fig, 7(a), we replaced ReLU activation with Softplus and retrained the network (with the ReLU network weights as initial weights). The resulting accuracy was 73%.

| Network Layers |
| --- |
| $3 \times 3$ conv. 96 ReLU |
| $3 \times 3$ conv. 96 ReLU |
| $3 \times 3$ conv. 96 Relu  Stride 2 |
| $3 \times 3$ conv. 192 ReLU |
| $3 \times 3$ conv. 192 ReLU |
| $3 \times 3$ conv. 192 Relu  Stride 2 |
| 1024 hidden sized feed forward |

## B   ADDITIONAL EXAMPLES OF FEATURE IMPORTANCE PERTURBATIONS

Here we provide three more examples from ImageNet.  For each example, all three methods of feature importance are attacked by random sign noise and our two targeted adversarial algorithms.

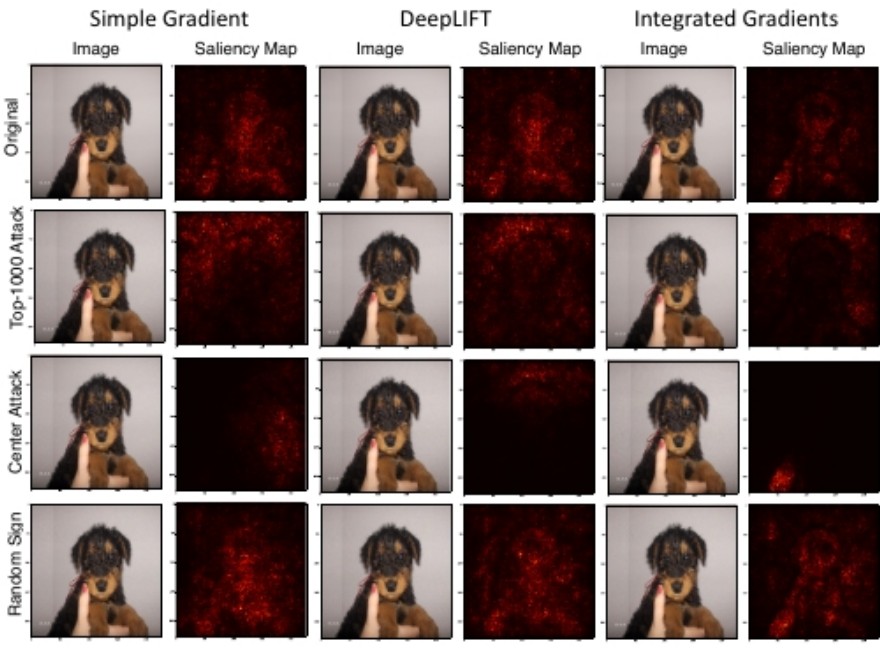

Figure 8: All of the images are classified as a *airedale*.

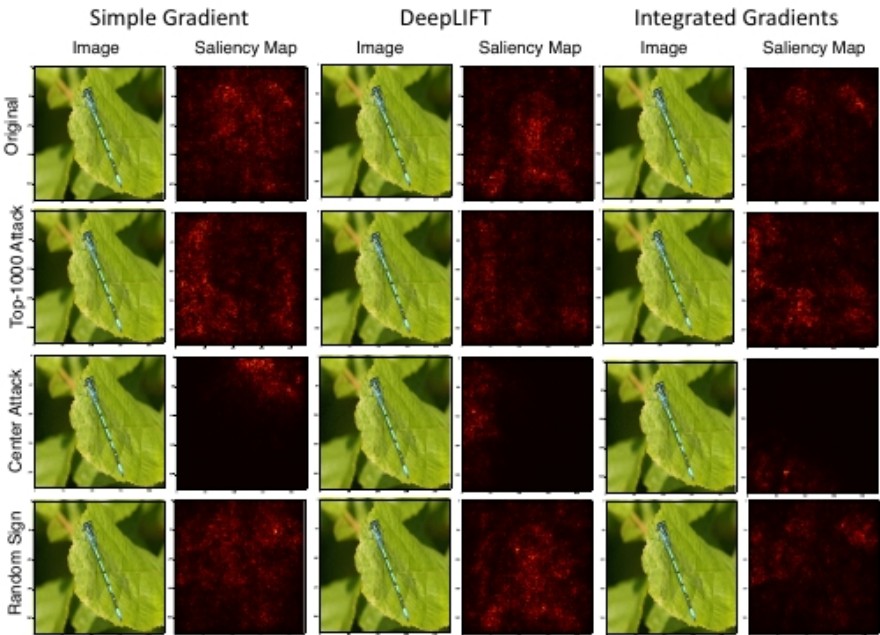

Figure 9: All of the images are classified as a *damselfly*.

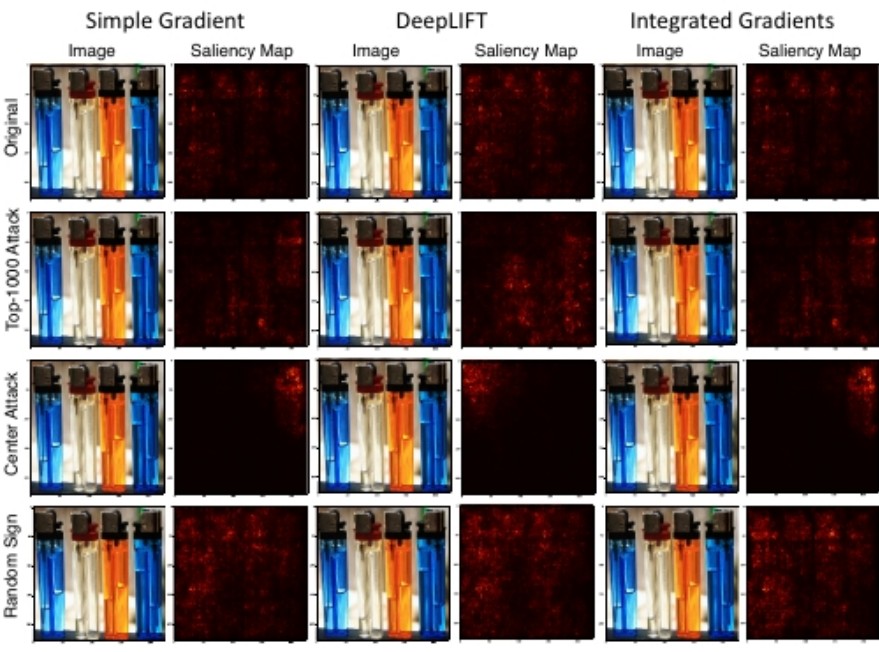

Figure 10: All of the images are classified as a *lighter*.

# C OBJECTIVE METRICS AND SUBJECTIVE CHANGE IN FEATURE IMPORTANCE MAPS

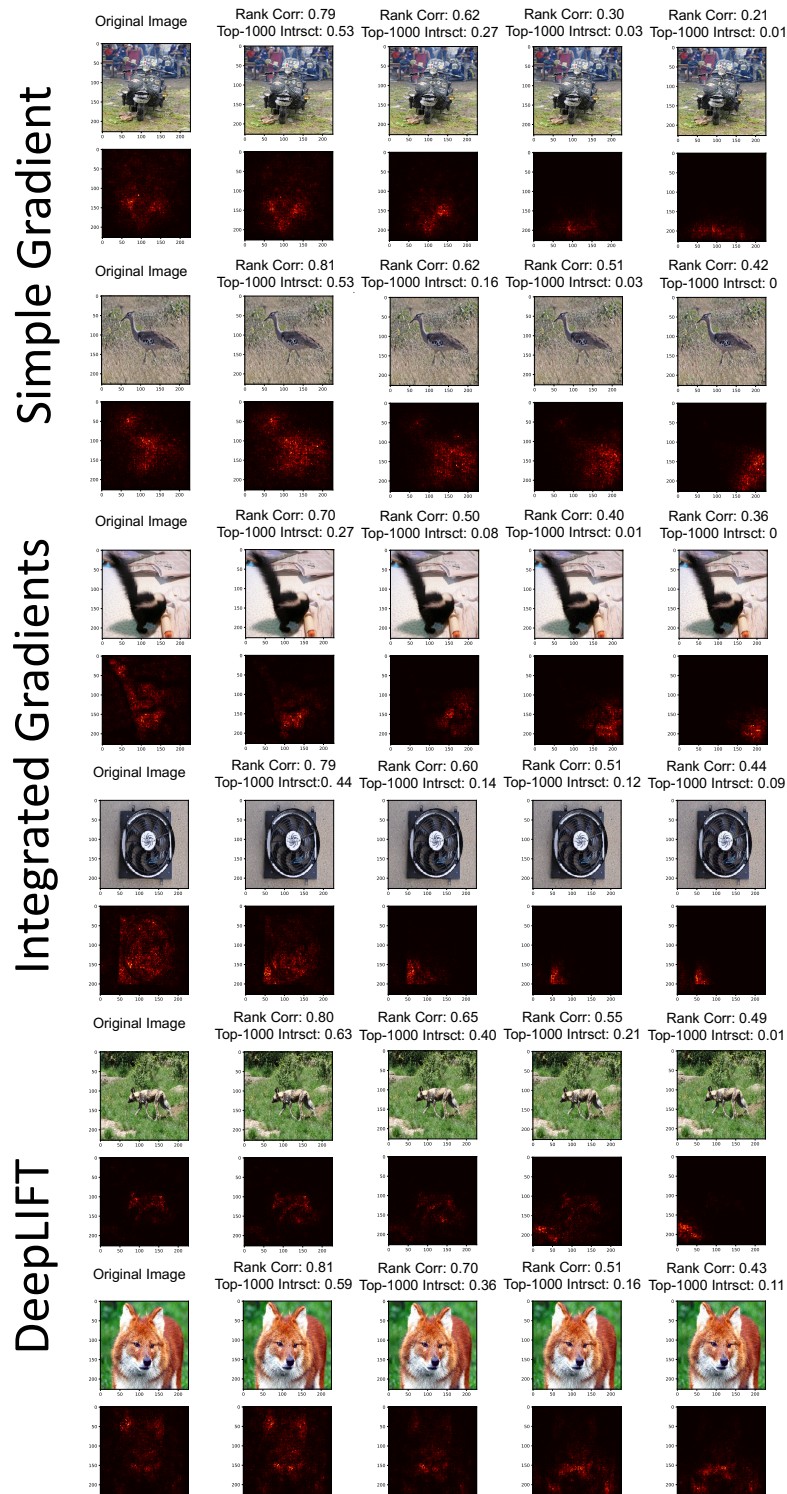

Figure 11: **Evaluation metrics vs subjective change in saliency maps** To have a better sense of how rank order correlation and top-1000 intersection metrics are related to changes in saliency maps, snapshots of the iterations of mass-center attack are depicted.

# D    MEASURING CENTER OF MASS MOVEMENT

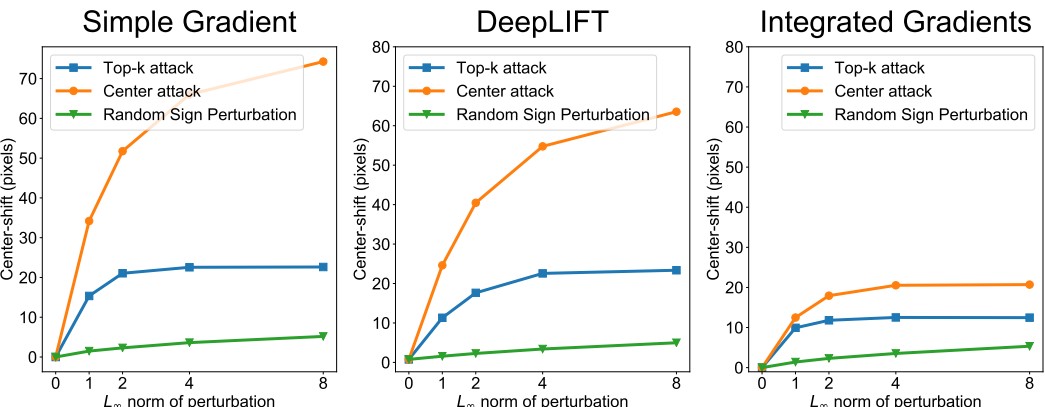

Figure 12: **Center-shift results for three feature importance methods on ImageNet**: As discussed in the paper, among our three measurements, center-shift measure was the most correlated measure with the subjective perception of change in saliency maps. The results in Appendix B also show that the center attack which resulted in largest average center-shift, also results in the most significant subjective change in saliency maps. Random sign perturbations, on the other side, did not substantially change the global shape of the saliency maps, though local pockets of saliency are sensitive. Just like rank correlation and top-1000 intersection measures, the integrated gradients method is the most robust method against adversarial attacks in the center-shift measure .

RESULTS FOR ADVERSARIAL ATTACKS AGAINST CIFAR-10 FEATURE IMPORTANCE METHODS

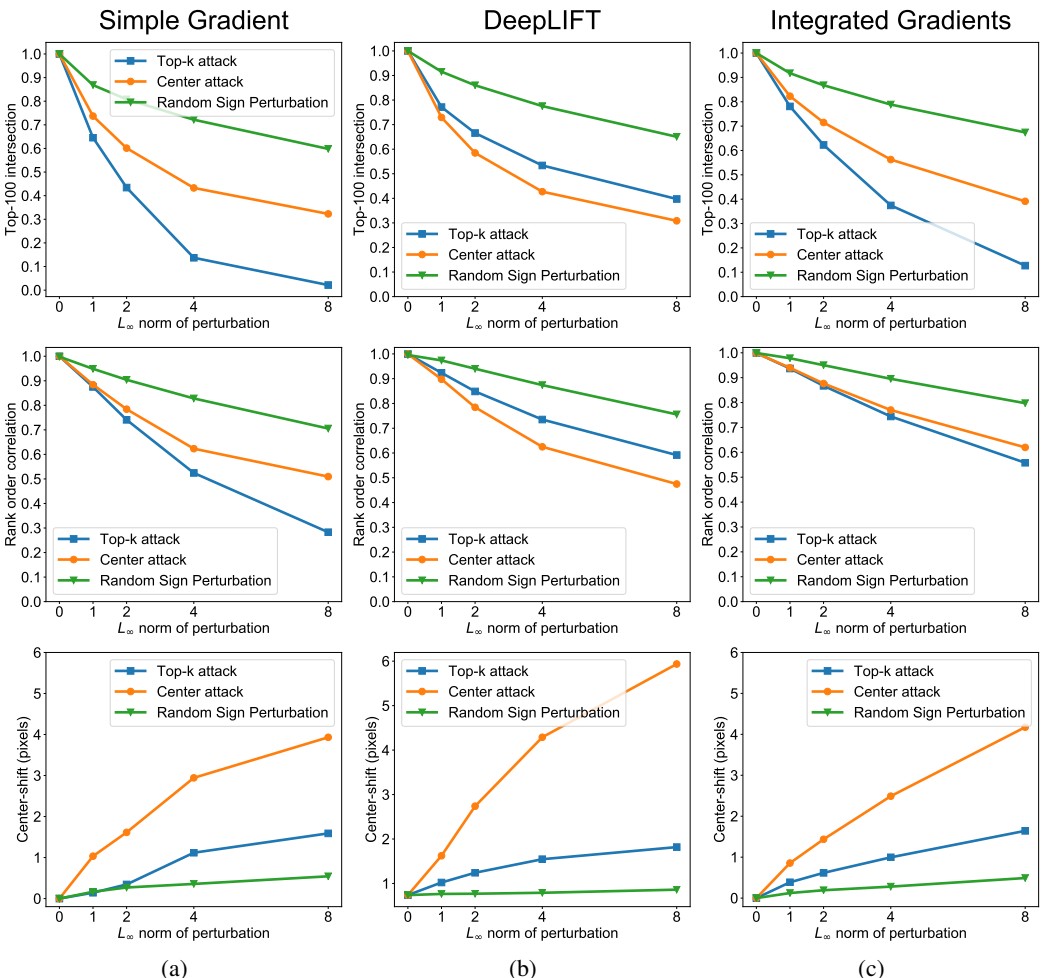

Figure 13: **Results for adversarial attacks against CIFAR10 feature importance methods**: For CIFAR10 the mass-center attack and top-k attack with k=100 achieve similar results for rank correlation and top-100 intersection measurements and both are stronger than random perturbations. Mass-center attack moves the center of mass more than two other perturbations. Among different feature importance methods, integrated gradients is more robust than the two other methods. Additionally, results for CIFAR10 show that images in this data set are more robust against adversarial attack compared to ImageNet images which agrees with our analysis that higher dimensional inputs are tend to be more fragile.

# E    ADDITIONAL EXAMPLES OF ADVERSARIAL ATTACKS ON INFLUENCE FUNCTIONS

In this appendix, we provide additional examples of the fragility of influence functions, analogous to Fig. 5.

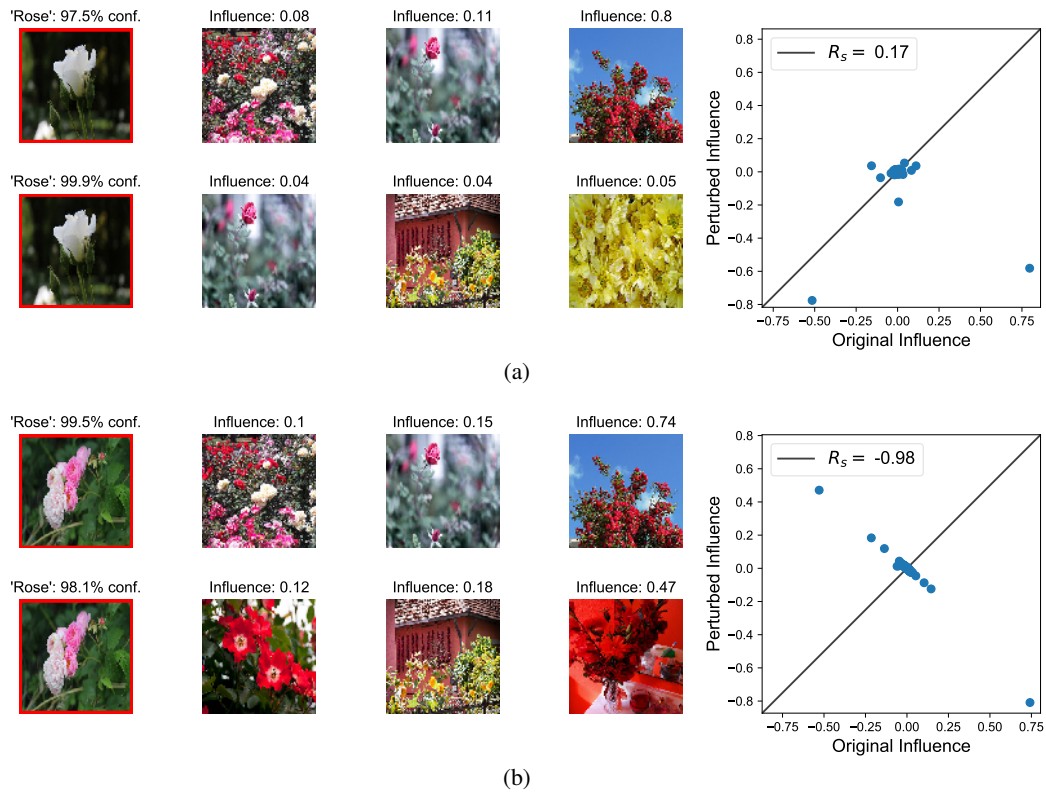

Figure 14: **Further examples of gradient-sign attacks on influence functions.** (a) Here we see a representative example of the most influential training images before and after a perturbation to the test image. The most influential image before the attack is one of the least influential afterwards. Overall, the influences of the training images before and after the attack are uncorrelated. (b) In this example, the perturbation has remarkably caused the training images to almost completely reverse in influence. Training images that had the most positive effect on prediction now have the most negative effects and the other way round.

## F DIMENSIONALITY-BASED EXPLANATION FOR FRAGILITY OF INFLUENCE FUNCTIONS

Here, we demonstrate that increasing the dimension of the input of a simple neural network increases the fragility of that network with respect to influence functions, analogous to the calculations carried out for importance-feature methods in Section 5. Recall that the influence of a training image $z_i = (\boldsymbol{x}_i, y_i)$ on a test image $z = (\boldsymbol{x}, y)$ is given by:

$$I(z_i, z) = - \underbrace{\nabla_\theta L(z, \hat{\theta})^\top}_{\text{dependent on } \boldsymbol{x}} \underbrace{H_{\hat{\theta}}^{-1} \nabla_\theta L(z_i, \hat{\theta})}_{\text{independent of } \boldsymbol{x}}. \tag{6}$$

We restrict our attention to the term in (6) that is dependent on $\boldsymbol{x}$, and denote it by $J \stackrel{\text{def}}{=} \nabla_\theta L$. $J$ represents the infinitesimal effect of each of the parameters in the network on the loss function evaluated at the test image.

Now, let us calculate the change in this term due to a small perturbation in $\boldsymbol{x} \to \boldsymbol{x} + \boldsymbol{\delta}$. The first-order approximation for the change in $J$ is equal to: $\nabla_{\boldsymbol{x}} J \cdot \boldsymbol{\delta} = \nabla_\theta \nabla_{\boldsymbol{x}} L \cdot \boldsymbol{\delta}$. In particular, for the $i^{\text{th}}$ parameter, $J_i$ changes by $(\nabla_\theta \nabla_{\boldsymbol{x}} L \cdot \boldsymbol{\delta})_i$ and furthermore, the relative change is $(\nabla_\theta \nabla_{\boldsymbol{x}} L \cdot \boldsymbol{\delta})_i / (\nabla_\theta L)_i$. For the simple network defined in Section 5, this evaluates to (replacing $\theta$ with $\boldsymbol{w}$ for consistency of notation):

$$\frac{(\boldsymbol{x}\boldsymbol{w}^\top \boldsymbol{\delta} g''(\boldsymbol{w}^\top \boldsymbol{x}))_i}{(\boldsymbol{x} g'(\boldsymbol{w}^\top \boldsymbol{x}))_i} = \frac{x_i \boldsymbol{w}^\top \boldsymbol{\delta} g''(\boldsymbol{w}^\top \boldsymbol{x})}{x_i g'(\boldsymbol{w}^\top \boldsymbol{x})} = \frac{\boldsymbol{w}^\top \boldsymbol{\delta} g''(\boldsymbol{w}^\top \boldsymbol{x})}{g'(\boldsymbol{w}^\top \boldsymbol{x})}, \tag{7}$$

where for simplicity, we have taken the loss to be $L = |y - g(\boldsymbol{w}^\top x)|$, making the derivatives easier to calculate. Furthermore, we have used $g'(\cdot)$ and $g''(\cdot)$ to refer to the first and second derivatives of $g(\cdot)$. Note that $g'(\boldsymbol{w}^\top \boldsymbol{x})$ and $g''(\boldsymbol{w}^\top \boldsymbol{x})$ do not scale with the dimensionality of $\boldsymbol{x}$ because $\boldsymbol{x}$ and $\boldsymbol{w}$ are generalized $L_2$-normalized due to data preprocessing and weight decay regularization.

However, if we choose $\boldsymbol{\delta} = \epsilon \text{sign}(\boldsymbol{w})$, then the relative change in the saliency grows with the dimension, since it is proportional to the $L_1$-norm of $\boldsymbol{w}$.

## G    ORTHOGONALITY OF STEEPEST DIRECTIONS OF CHANGE IN SCORE AND FEATURE IMPORTANCE FUNCTIONS IN A SIMPLE TWO-LAYER NETWORK

Consider a two layer neural network with activation function $g(\cdot)$, input $\boldsymbol{x} \in \mathbb{R}^d$, hidden vector $\boldsymbol{u} \in \mathbb{R}^h$ , and score function $S$), we have:

$$S = \boldsymbol{v} \cdot \boldsymbol{u} = \sum_{j=1}^{h} v_j u_j$$

$$\boldsymbol{u} = g(W^T \boldsymbol{x}) \rightarrow u_j = \boldsymbol{w}_j.\boldsymbol{x}$$

where $\boldsymbol{w}_j = ||\boldsymbol{w}_j||_2 \hat{\boldsymbol{w}}_j$. We have:

$$\nabla_{\boldsymbol{x}} S = \sum_{j=1}^{h} v_j \nabla_{\boldsymbol{x}} u_j = \sum_{j=1}^{h} v_j g'(\boldsymbol{w}_j.\boldsymbol{x}) \boldsymbol{w}_j$$

$$\nabla_{\boldsymbol{x}}^2 S = \sum_{j=1}^{h} v_j \nabla_{\boldsymbol{x}}^2 u_j = \sum_{j=1}^{h} v_j g''(\boldsymbol{w}_j.\boldsymbol{x}) \boldsymbol{w}_j^T \boldsymbol{w}_j$$

Now for an input sample $\boldsymbol{x}$ perturbation $\boldsymbol{\delta}$, for the change in feature importance:

$$\nabla_{\boldsymbol{x}} S(\boldsymbol{x} + \boldsymbol{\delta}) - \nabla_{\boldsymbol{x}} S(\boldsymbol{x}) \approx \nabla_{\boldsymbol{x}}^2 S \cdot \boldsymbol{\delta}$$

which is equal to:

$$\sum_{j=1}^{h} v_j g''(\boldsymbol{w}_j.\boldsymbol{x})(\boldsymbol{w}_j \cdot \boldsymbol{\delta}) \boldsymbol{w}_j$$

We further assume that the input is high-dimensional so that $h < d$ and for $i \neq j$ we have $\boldsymbol{w}_j \cdot \boldsymbol{w}_i = 0$. For maximizing the $\ell_2$ norm of saliency difference we have the following perturbation direction:

$$\boldsymbol{\delta}_m = \text{argmax}_{||\boldsymbol{\delta}||=1} ||\nabla_{\boldsymbol{x}} S(\boldsymbol{x} + \boldsymbol{\delta}) - \nabla_{\boldsymbol{x}} S(\boldsymbol{x})|| = \hat{\boldsymbol{w}}_k$$

where:

$$k = \text{argmax}|v_j g''(\boldsymbol{w}_j.\boldsymbol{x})| \times ||\boldsymbol{w}_k||_2^2$$

comparing which to the direction of feature importance:

$$\frac{\nabla_{\boldsymbol{x}} S(\boldsymbol{x})}{||\nabla_{\boldsymbol{x}} S(\boldsymbol{x})||_2} = \sum_{i=1}^{h} \frac{v_j g'(\boldsymbol{w}_i \cdot \boldsymbol{x}) ||\boldsymbol{w}_i||_2}{(\sum_{j=1}^{h} v_j g'(\boldsymbol{w}_j \cdot \boldsymbol{x}) ||\boldsymbol{w}_j||_2)^2} \hat{\boldsymbol{w}}_i$$

we conclude that the two directions are not parallel unless $g'(.) = g''(.)$ which is not the case for many activation functions like Softplus, Sigmoid, etc.

## H   DESIGNING INTERPRETABILITY-ROBUST NETWORKS

The analyses and experiments in this paper have demonstrated that small perturbations in the input layers of deep neural networks can have large changes in the interpretations. This is analogous to classical adversarial examples, whereby small perturbations in the input produce large changes in the *prediction*. In that setting, it has been proposed that the Lipschitz constant of the network be constrained during training to limit the effect of adversarial perturbations (Szegedy et al., 2013). This has found some empirical success (Cisse et al., 2017).

Here, we propose an analogous method to upper-bound the change in interpretability of a neural network as a result of perturbations to the input. Specifically, consider a network with $K$ layers, which takes as input a data point we denote as $y_0$. The output of the $i^{\text{th}}$ layer is given by $y_{i+1} = f_i(y_i)$ for $i = 0, 1 \ldots K - 1$. We define $S \overset{\text{def}}{=} f_{K-1}(f_{K-2}(\ldots f_0(y_0) \ldots))$ to be the output (e.g. score for the correct class) of our network, and we are interested in designing a network whose gradient $S' = \nabla_{y_0} S$ is relatively insensitive to perturbations in the input, as this corresponds to a network whose feature importances are robust.

A natural quantity to consider is the Lipschitz constant of $S'$ with respect to $y_0$. By the chain rule, the Lipschitz constant of $S'$ is

$$\mathcal{L}(S') = \mathcal{L}(\frac{\delta y_k}{\delta y_{k-1}}) \ldots \mathcal{L}(\frac{\delta y_1}{\delta y_0}) \tag{8}$$

Now consider the function $f_i(\cdot)$, which maps $y_i$ to $y_{i+1}$. In the simple case of the fully-connected network, which we consider here, $f_i(y_i) = g_i(W_i y_i)$, where $g_i$ is a non-linearity and $W_i$ are the trained weights for that layer. Thus, the Lipschitz constant of the $i^{\text{th}}$ partial derivative in (8) is the Lipschitz constant of

$$\frac{\delta f_i}{\delta y_{i-1}} = W_i g_i'(W_i y_{i-1}),$$

which is upper-bounded by $||W_i||^2 \cdot \mathcal{L}(g_i'(\cdot))$, where $||W||$ denotes the operator norm of $W$ (its largest singular value)[5]. This suggests that a conservative upper ceiling for (8) is

$$\mathcal{L}(S') \leq \prod_{i=0}^{K-1} ||W_i||^2 \mathcal{L}(g_i'(\cdot)) \tag{9}$$

Because the Lipschitz constant of the non-linearities $g_i'(\cdot)$ are fixed, this result suggests that a regularization based on the operator norms of the weights $W_i$ may allow us to train networks that are robust to attacks on feature importance. The calculations in this Appendix section is meant to be suggestive rather than conclusive, since in practice the Lipschitz bounds are rarely tight.

---

[5]this bound follows from the fact that the Lipschitz constant of the composition of two functions is the product of their Lipschitz constants, and the Lipschitz constant of the product of two functions is also the product of their Lipschitz constants.

