# OpenReview forum: "INTERPRETATION OF NEURAL NETWORK IS FRAGILE"
_ICLR.cc/2018/Conference — Reject_

### Official Review · AnonReviewer1 · 2017-11-21
**Interpretation of Neural Network is Fragile**

**Rating:** 6
**Confidence:** 2

**Review:**

The authors study cases where interpretation of deep learning predictions is extremely fragile. They systematically characterize the fragility of several widely-used feature-importance interpretation methods. In general, questioning the reliability of the visualization techniques is interesting. Regarding the technical details, the reviewer has the following comments:

- What's the limitation of this attack method?

- How reliable are the interpretations?

- The authors use spearman's rank order correlation and Top-k intersection as metrics for interpretation similarity.

- Understanding whether influence functions provide meaningful explanations is very important and challenging problem in medical imaging applications. The authors showed that across the test images, they were able to perturb the ordering of the training image influences. I am wondering how this will be used and evaluated in medical imaging setting.

---

> ### Author Response · Authors · 2017-12-14
> **Thanks.**
>
> Thank you for the review and feedback. The main contribution of our paper is to systematically demonstrate for the first time that interpretation of neural networks are fragile to attacks. This is an important topic and your questions raise interesting future research directions.
>
> 1. The limitation of the attack method is a very interesting research direction. The attacks that we designed in our paper are all white-box attacks that need to know the NN model. Our next question to answer would be the dangers of interpretations attacks in the black-box setting without access to the model.
>
> 2.  The reviewer asks how reliable are the interpretation methods. Although these methods are widely used(e.g. Quang and Xie 17, Kelly and Reshev 17), there is not a unified definition of reliability that has been investigated and it is an active area of research(Doshi-Velez & Kim, 2017). One of the contributions of our work is to systematically compare the robustness of the interpretations generated by these different methods. Our work shows that is possible to change regions of high saliency through careful perturbations of the test images (see Fig. 2). So the methods are correctly identifying new interpretations, but these interpretations disagree with human notions of what part of an image is most related to interpretation.
>
> 3. As the reviewer mentions, we defined metrics (rank correlation and top-k intersection, and also center shift metric in Appendix D) to compare the interpretations of two different images. In order to make it clear how these metrics correspond to intuitive notions of stability, we have included a new figure, Figure 3, and a new appendix, Appendix C, which provides an example of how rank correlations and top-k intersection change as randomly sampled validation images are adversarially perturbed.
>
> 4 .We agree that medical case is one of the most important problems for the application influence functions and one way to evaluate the perturbations is to look at the concordance with human studies.

---

### Official Review · AnonReviewer2 · 2017-11-27
**The key point of the paper is that it is possible to get different salience maps for interpretability while retaining correct labeling.**

**Rating:** 4
**Confidence:** 4

**Review:**

The key observation is that it is possible to generate adversarial perturbations wherein the behavior of feature importance methods (e.g. simple gradient method (Simonyan et al, 2013), integrated gradient (Sundararajan et al, 2017), and DeepLIFT ( Shrikumar et al, 2016) ) have large variation while predicting same output.    Thus the authors claim that one has to be careful about using feature importance maps.

Pro:  The paper raises an interesting point about the stability of feature importance maps generated by gradient based schemes.

Cons:
The main problem I have with the paper is that there is no precise definition of what constitutes the stable feature importance map.  The examples in the paper seem to be cherry picked to illustrate dramatic effects.   The experimental protocol used does not provide enough information of the variability of the salience maps shown around small perturbations of adversarial inputs. The paper would benefit from more systematic experimentation and a better definition of what authors believe are important attributes of stability of human interpretability of neural net behavior.

---

> ### Author Response · Authors · 2017-12-14
> **We have a precise definition of fragility and support it with systematic experiments.**
>
> Thank you for the review and feedback. Here, we address the points made in the review as well as describe changes to the original submission to incorporate the reviewer’s feedback.
>
> Our paper proposes a precise definition of what it means for an interpretation to be fragile. As we stated in the abstract, our definition is “two perceptively indistinguishable inputs with the same predicted label can be assigned very different interpretations”.  A stable feature map is one that is not fragile by this definition. We have included additional discussions of our definitions in the Our Contributions section to clarify any question. Moreover, we proposed two clear metrics, rank correlation and top-K intersect, to quantify exactly how different two interpretations are. We have also included a new figure, Figure 3, and a new appendix, Appendix C, which provides an example of how rank correlations and top-k intersection change as randomly sampled validation images are adversarially perturbed.
>
> The examples in Figure 1 are representative of how interpretations can be attacked by our perturbations. We have released our code at [https://goo.gl/6usSEk] and the reviewer can verify for him/herself that our attacks are reproducible and consistent for ImageNET and CIFAR10. Moreover, our experiments on ImageNET and CIFAR10 does systematically support that the interpretations are fragile by our definitions (Figs 4, 5 in the main text and Figs 12, 13 in the Appendix).
>
> Could you please let us know if you have any more questions regarding the paper or if there are specific experiments that you’d like to see? We’d like to engage in a dialogue until we resolve all of your questions.

---

### Official Review · AnonReviewer3 · 2017-11-27
**Interesting work, but wrong conclusions**

**Rating:** 5
**Confidence:** 5

**Review:**

The paper shows that interpretations for DNN decisions, e.g. computed by methods such as sensitivity analysis or DeepLift, are fragile: Visually (to a human) inperceptibly different image cause greatly different explanations (and also to an extent different classifier outputs). The authors perturb input images and create explanations using different methods. Even though the image is inperceptibly different to a human observer, the authors observe large changes in the heatmaps visualizing the explanation maps. This is true even for random perturbations.

The images have been modified wrt. to some noise, such that they deviate from the natural statistics for images of that kind. Since the explanation algorithms investigated in this papers merely react to the interactions of the model to the input and thus are unsupervised processes in nature, the explanation methods merely show the model's reaction to the change.
For one, the model itself reacts to the perturbation, which can be measured by the (considarbly) increased class probability. Since the prediction score is given in probabilty values, the reviewer assumes the final layer of the model is a SoftMax activation. In order to see change in the softmax output score, especially if the already dominant prediction score is further increased, a lot of change has to happen to the outputs of the layer serving as input to the SoftMax layer.

It can thus be expected, that the input- and class specific explanations change as well, to an also not so small extent. The explanation maps mirror for the considered methods the model's reaction to the input. They are thus not meaningless, but are a measure to model reaction instead of an independent process. The excellent Figure 2 supports this point. Not the interpretation itself is fragile, but the model.
Adding a small delta to the sample x shifts its position in data space, completely altering the prediction rule applied by the model due to the change in proximity to another section of the decision hyperplane. The fragility of DNN models to marginally perturbed inputs themselves is well known.
This especially true for adversial perturbations, which have been used as test cases in this work. The explanation methods are expected to highlight highly important areas in an image, which have been targetet by these perturbation approaches.

The authors give an example of an adversary manipulating the input in order to draw the activation to specific features to draw confusing/malignant explanation maps. In a settig of model verification, the explanation via heatmaps is exactly what one wants to have: If tiny change to the image causes lots of change to the prediction (and explanation) we can visualize the instability of the model not the explanation method.
Further do targeted perturbations not show the fragility of explanation methods, but rather that the models actually find what is important to the model. It can be expected, that after a change to these parts of the input, the model will decide differently, albeit coming to the same conclusion (in terms of predicted class membership), which reflects in the explanation map computed for the perturbed input.

Further remarks:
It would be interesting to see the size and position of the center of mass attacks in the appendix. The reviewer closely follows and is experienced with various explanation methods, their application and the quality of the expected explanations. The reviewer is therefore surprised by the poor quality and lack of structure in the maps obtained from the DeepLift method. Can bugs and suboptimal configurations be ruled out during the experiments? The DeepLift explanations are almost as noisy as the ones obtained for Sensitivity Analysis (i.e. the gradient at the input point). However, recent work (e.g. Samek et al., IEEE TNNLS, 2017 or Montavon et al., Digital Signal Processing, 2017) showed that decomposition-based methods (such as DeepLift) provide less noisy explanations than Sensitivity Analysis.

Have the authors considered training the net with small random perturbations added to the samples, to compare the "vanilla" model to the more robust one, which has seen noisy samples, and compared explanations?
Why not train (finetune) the considered models using softplus activations instead of exchanging activation nodes?
Appendix B: Heatmaps through the different stages of perturbation should be normalized using a common factor, not individually, in order to better reflect the change in the explanation

Conclusion:
The paper follows an interesting approach, but ultimately takes the wrong view point:
The authors try to attribute fragility to explaining methods, which visualize/measure the reaction of the model to the perturbed inputs. A major rework should be considered.

---

> ### Author Response · Authors · 2017-12-14
> **Our conclusions are the same as yours.**
>
> Thank you for the thorough review and feedback. Our conclusions are actually exactly the same as yours! Our main contribution is to show that the interpretation (e.g. the saliency map of an image) can be significantly perturbed by adversarial attacks; we do not claim that the interpretation method (e.g. DeepLift) is broken by the attacks. This completely agrees with your point.
>
> Our operational definition that an interpretation is fragile is “two perceptively indistinguishable inputs with the same predicted label can be assigned very different interpretations”. All of our experiments support our conclusion that interpretations can be adversarially attacked by this definition. We used this definition because it’s analogous to the fragility notion used in the broader adversarial attack and ML security community.  We do not claim in the paper that the interpretation method itself is broken by our attacks. This is a subtle and important distinction. We have added a new paragraph in the Conclusion  section to clarify this:
>
> “Our results demonstrate that the \emph{interpretations} (e.g. saliency maps) are vulnerable to perturbations, but this does not imply that the \emph{interpretation methods} are broken by the perturbations. This is a subtle but important distinction. Methods such as saliency measure the infinitesimal sensitivity of the neural network at a particular input $\xb$. After a perturbation, the input has changed to $\tilde{\xb} = \xb + \deltb$, and the salency now measures the  sensitivity at the perturbed input. The saliency \emph{correctly} captures the infinitesimal sensitivity at the two inputs; it's doing what it is supposed to do. The fact that the two resulting saliency maps are very different is fundamentally due to the network itself being fragile to such perturbations, as we illustrate with Fig.~\ref{fig:concept}.”
>
> You asked about the quality of our DeepLIFT saliency maps. We implemented DeepLIFT with rescale rule as described by the original authors [https://arxiv.org/abs/1704.02685 ]. We have released the code for our implementation at [https://goo.gl/6usSEk]. We have checked the code several times and we also closely work with the the lab developing DeepLIFT in the area of interpretability.
>
> Regarding your remark that heat-maps should be normalized using a common factor. It’s important to notice that the measures used for comparing the original and perturbed saliency maps (top-K intersection and rank order correlation) are independent from the normalizing scheme.
>
> Regarding your suggestion a training strategy to make networks more robust to such adversarial examples, and suggested retraining a network using softplus activations. These are very helpful suggestions that we will consider for future direction.
>
> Does this help to address your question? We believe that our conclusions are the same as yours. Please let us know if the new discussions we’ve added to the revision clarify this confusion. The key contribution of our work is to extend adversarial attacks to interpretations for the first time, and this raises interesting security questions given how important interpretations are. Please let us know if there are additional analysis you’d like to see. We hope to engage in a dialogue until we can resolve all of your concerns.

---

### Public Comment · (anonymous) · 2017-11-06
**Reliability of Interpretations on Original Images**

I was wondering how reliable are the interpretations? i.e., for what fraction of original images do they generate a "meaningful" interpretation?

---

> ### Author Response · Authors · 2017-11-06
> **Reliability of Interpretations on Original Images**
>
> Very good question.
>     To have a sense of reliability of saliency methods, one can argue with both subjective and objective measures. The most prevalent objective measure in the literature has been weakly-supervised object localization which is basically trying to localize the classified object using the most salient input dimensions(pixels). Simonyan et al discussed this measure in their original work for the simple gradient method and reported less than 50% error. DeepLIFT and Integrated Gradients methods have not been yet applied to the task of localization to the best of our knowledge.
>     As a subjective comment, however, I have to say that all of the three mentioned feature importance methods are successful (nearly all the time) in pointing to the region of important pixels (in other words the region of image containing the classified object); what makes them different in performance, is how noisy the saliency map is. In other words, how many non-important pixels(subjectively) are pointed out by the feature importance method to be important or how many missing important pixels one can detect in the salient part of the heat map. It could be said that Integrated Gradients results in the best subjective saliency map and also DeepLIFT has acceptable performance. Examples of both could be found in https://github.com/ankurtaly/Integrated-Gradients and https://github.com/kundajelab/deeplift, respectively. Also, the recent "SmoothGrad: removing noise by adding noise" paper has tried to solve the problem of noisy saliency maps and they have reported convincing results.
>     Understanding whether influence functions provide meaningful explanations is more challenging since training examples can influence the prediction of a test image in different ways. For example, a very similar-looking training image from the same class may help the classifier identify the test image, but so can a very different-looking training image from a different class. As such, we find that the training examples that are identified as the most helpful by influence functions are not generally the same as those images that are visually similar to the test image. However, regardless of the meaningfulness of influence functions on a particular test image, we show that across the test images, we are able to perturb the ordering of the training image influences.

---

> > ### Public Comment · (anonymous) · 2017-11-06
> > **Interpretation on Training vs Test Data**
> >
> > Thanks for your thorough reply. Another question: regarding subjective measure of interpretation, do feature importance methods perform similarly on training and test data?

---

> > > ### Author Response · Authors · 2017-11-10
> > > **Re: Interpretation on Training vs Test Data**
> > >
> > > Generally speaking, we find similar performance between interpretability on training images and test images. Of course, if a training image is used at test time, influence functions will return the training image itself as the most influential image.

---

> > > > ### Public Comment · (anonymous) · 2017-11-10
> > > > **Feature importance methods**
> > > >
> > > > My question is more about identifying important features rather than finding influential training images. I wonder if feature importance methods produce similar good results on training and test images?

---

### Public Comment · ~yang_zhang1 · 2017-11-16
**This attack method is not applicable to any network with pooling layers**

To apply the attack method, we cannot avoid calculating the gradient of the saliency map w.r.t to the input. However, notice that the saliency map itself is a gradient of the maxlogit w.r.t the input. The gradient for a neural network is usually calculated by using the backpropagation, which depends on the forward pass. So, for any network with pooling layers, the gradient of the saliency map w.r.t the input is not calculable, because the pooling layers, when you calculate the gradient for the second time, are not differentiable. In the paper, you only point out that for the widely used activation function Relu, in order to avoid the zero gradient problem, we can replace it with a softplus activation.

One possible way to solve this problem is to stop the gradient of D w.r.t the input x at the saliency map level. However, this fix doesn't make sense mathematically, though it might can still be an attack method.

Considering that most of the networks will contain pooling layers, I think the application of this attack method is limited. But questioning the reliability of the visualization techniques is a good point in general.

---

> ### Author Response · Authors · 2017-11-17
> **The method is applicable to a networks with pooling layers**
>
> The interpretation attacks in our paper do work for NNs with pooling layers. In fact, our experimental results on ImageNet (Figs 1 and 3) were all produced for Sqeezenet, which has maxpooling and average pooling layers. The reason this works is as follows: when computing the attack direction, we replace ReLU with Softplus so we have non-zero second gradients. Max-pooling simply picks out a particular neuron to pass through, so it will have non-zero second order gradients as long as the activation has non-zero second gradient  (except on a set of measure zero).  The second order chain rule has two terms, one of which, is zero as the second order gradient of maxpooling is zero. The other term is non-zero as long as the activation functions of the network have non-zero second order gradients. For more clarification please look at Faà di Bruno's formula for second order derivative.  (You can empirically confirm it by building a simple network in Tensroflow with softplus activation and a maxpool layer and take the second order gradient).
> Note that we use this Softplus network only for the purpose of finding an attack direction, we still compute saliency with respect to the original ReLU network, as we discussed in the paper. This means that our attacks can be applied to any ReLU network with or without pooling.
> You are correct that the main message of our paper is that reliability of neural network interpretation is fragile, and we developed approaches to systematically quantify this effect for the first time. We expect that there are many ideas to improve upon the specific attacks we proposed, and this opens up an interesting research direction.

---

> > ### Public Comment · ~yang_zhang1 · 2017-11-17
> > **Is "tf.gradients(tf.gradients(maxlogit, input), input)" legal?**
> >
> > Thanks for your reply.
> >
> > I think you would agree that to apply your attack algorithm in the paper, you cannot avoid calculating the above gradient in the title, i.e the gradient of the saliency map w.r.t the input. However, this expression itself is not legal. Notice that the first gradient inside is actually the saliency map. We do can calculate it because we know the forward pass. However, for the gradient outside, we will have a problem because now we don't know what's the gradient for the pooling layers. I don't think Softplus will solve this issue.
> >
> > I was wondering if you could release your code. Maybe you have a smart way to get around based on some math derivations.
> >
> > Thank you again.

---

> > > ### Author Response · Authors · 2017-11-17
> > > **Straight Forward Implementation**
> > >
> > > We will release our code soon to show how the algorithm is implemented. The gradients can be implemented in Tensorflow like we described in the paper.

---

### Public Comment · (anonymous) · 2017-11-22
**With four parameters I can fit an elephant, and with five I can make him wiggle his trunk - John von Neumann**

Dear Authors,

1) In Figure 8 the three saliency maps differ from each other. This implies that at least 2 of the saliency maps are incorrect/irrelevant to this problem (may be all the 3 are).
    To corroborate the above point:
    a) In Figure 9, the DEEPLift saliency map differs from the other two
    b) In Figure 7, the Integrated saliency map differs from the other two.
    From this, one would think that the simple gradient method is the most reliable of the three methods but Section 2.1 contradictingly states that (Shrikumar, Sundarajan) "simple gradient method" has drawbacks.

2) In Figure 1, the confidence of the output of the network has increased in all the three of your examples. Is this a general phenomenon or are these hand picked examples?

3) The introduction stresses equally on
    a) "importance of individual features or training examples is highly fragile to even small random perturbations to the input image".
    b) "we show how targeted perturbations can lead to different global interpretations".
    However, surprisingly Figure 1 doesn't have any examples based on random perturbation attack.
    On further searching, I was able to find an example in the appendix, where the changes in the saliency map due to random pertubation is nothing compared to changes in the saliency map due to attacked perturbation. I feel the case for random permutation attack is over-emphasized.

4) In section 2.2, each test image has a vector v of size |train_images|, where the element v[i] is the importance of training image i for the classification of the test image. For high dimensional vectors v1, v2, the spearman rank correlation is very prone to small noise.
To overcome this, the most 'natural' way to quantify the similarity between two different test images would be to take a dot product of their corresponding vectors v1 and v2. One could also using other similarity measures such as 2 -d(v1,v2), where d(v1,v2) is a distance metric designed to overcome the noise in high dimensions.

5) I was not able to find the public code of the experiments, even though this paper heavily relies on the experiments. Is the code public?

Thanks

---

> ### Author Response · Authors · 2017-11-22
> **Yes JvN would have agreed that the interpretation is fragile.**
>
> Thanks for your interest.
>
> All three saliency methods, including the simple gradient method, have drawbacks and are fragile. The best way to see this is the systematic experiments in Fig. 3.
>
> One of our findings is that, in general, perturbations can substantially change the interpretation (saliency) without changing the predicted label. In the systematic experiments of Sec. 4, ALL of the perturbations preserved the original label, and most of them led to very small changes in the prediction confidence.
>
> Our results show that our targeted perturbations are much more effective than random perturbations. We also demonstrate that the saliency of individual features (e.g. particular pixels) is fragile to even random perturbations. Fig. 3 top row shows that there is a ~80% turnover in the top features under random perturbation. These two results are complementary.
>
> Yes, we have used other similarity metrics such as intersection of the top features, L2 distance, etc. and the same results hold.
>
> As is common for most of the ICLR submissions, we plan make all the code public soon.

---

> > ### Public Comment · (anonymous) · 2017-11-22
> > **Deflected answers.**
> >
> > 1) "All three saliency methods, including the simple gradient method, have drawbacks and are unstable"
> >            Thus the conclusion "Interpretation of Neural networks is fragile" is based on unstable metrics and hence it requires much more evidence (stable metrics) than shown in the paper.
> >
> > 2) "most of them led to very small changes in the prediction confidence"
> >             Thus the figure 1 is not a good representative of the experimental results (since 1.a, 1.c have a huge change in their prediction confidence)
> >
> > 3) Sure.  So the authors agree that the statement "importance of individual features or training examples is highly fragile to even small random perturbations to the input image" is after all over-emphasized. As figure 3 illustrates, for small perturbations L_\infty = 1, random perturbations are around  3-10 times weaker than adversarial perturbations.
> >
> > 4) Why was spearman ranking chosen over L2? Can you share results for other distance measures anonymously? Many other submissions in ICLR do this.
> >
> > 5) Sure.

---

### Public Comment · (anonymous) · 2017-11-27
**Interesting observation but the conclusions drawn from the observation are incorrect.**

The paper describes an interesting effect but it attributes the observation incorrectly to interpretability method.
Therefore the title and conclusions drawn in the paper are not correct.

To be precise the paper claims that interpretability methods are fragile and that the experiments support this.
To claim this it is necessary to have two aspects.
A) A change is made such that the network is explained in a different way
B) This change should be controlled such that we are guaranteed that the explanation MUST be the same.
In this paper effect (A) is present, however part (B) is NOT present since the function the network computes changes.
If the network makes a different computation it is possible that the explanation needs to change as well.
Therefore the paper does not show that interpretation is fragile.

To understand why point (B) is missing we have to look at what is done and what is being interpreted.

It is shown that it is possible to imperceptibly change an image such that the gradient w.r.t. to the input changes, but without changing the classifier’s output.
Since most interpretability methods are based on the gradient this changes their interpretation.
This is where the paper makes a mistake by ignoring the meaning of the gradient or saliency map (as defined by Simonyan).

As noted in the Saliency Map paper by Simonyan, the gradient of the output (logit) w.r.t. the input
is a local linear approximation of the network’s function in the neighborhood of the datapoint, i.e. a Taylor series according to said paper.
For relu and max-pooling networks this approximation is actually exact given correct treatment of the bias.
Now when the gradient changes after an input change, this shows that the network computes a different function.

The approach in this paper assures that the most probable class does not change.
It does not control how this decision was made.
Since the function might be different, the explanation should not necessarily be the same.
It might very well be possible that the network’s decision needs to be interpreted differently now.
Intuitively, this can be understood by realizing that for many problems the same decision can be made based on different subsets of evidence.

Since there is no certainty that the decision was made in an identical manner for the original datapoint and the modified one,
 It is NOT possible to conclude that interpretability methods are fragile.

Additionally, the theoretical analysis also make an incorrect claim.
The paper states that a logistic regression network is susceptible to the adversarial attack on the interpretability.
This is not correct.
In the case that the logistic model (two or multi-class) is analyzed from the logit (as is done commonly in the papers by Bach et al, Simonyan et al ,…..) the gradient w.r.t. the input is always the weight vector.
In a two class setting where a sigmoid is added, the gradient becomes a scaled version of the weight vector.
For this reason interpretability is not fragile here either.
Therefore at least a multi class logistic regression is needed and the methods have to be applied to the output post-softmax.
Please not that this is not how the original papers describe the approach how to use these methods.

To conclude, I think the manuscript shows an interesting effect but it draws the wrong conclusions.
The fact that the network’s function can change drastically due to a small change in the input without affecting the prediction is surprising.
This is an effect similar to the original observation of adversarial examples.
However it does NOT show fragility of interpretability maps.
For this reason, I believe that the paper needs to be updated before it can be accepted.
The observed effect does warrant further investigation and is an interesting finding.

---

> ### Author Response · Authors · 2017-11-27
> **Our operational definition of fragility**
>
> Thanks for your comment.
>
> Our operational definition of fragility is that the interpretation is fragile if, given a fixed network, two similar inputs with the same prediction have very different ‘interpretations’. All of our results support the claim that interpretations are fragile by this metric.
>
> It seems like you are basically questioning whether this is a good metric of fragility and you might have a different metric in mind. We think it’s certainly interesting to explore other metrics, and our paper opens the door for that. There are two main motivations for our metric/definition of fragility.
>
> 1. Our definition is well-motivated by practice. Suppose you have a pathology image that is predicted to be a malignant tumor, as an example. The clinician might then use some saliency map to interpret which part of the image is the most informative. The reliability of this interpretation exactly maps onto our fragility metric: the input image always have measurement errors and our fragility metric quantifies whether the same parts of the image would light up as informative across different measurement errors. This motivates defining fragility as we do.
>
> 2.  Our definition is consistent with other notions of fragility to adversarial attacks.
>
> You said that the perturbation to the input changes the function. Yes, that’s exactly why the interpretation is fragile by our metric.

---

### Decision · Program_Chairs · 2018-01-29
**ICLR 2018 Conference Acceptance Decision**

**Decision:**

Reject

**Comment:**

The paper tries to show that many of the state-of-the-art interpretability methods are brittle and do not provide consistent stable explanations. The authors show this by perturbing (even randomly) the inputs so that the differences are imperceptible to a human observer but the interpretability methods provide completely different explanations. Although the output class is maintained before and after the perturbation it is not clear to me or the reviewers why one shouldn't have different explanations. The difference in explanations can be attributed to the fragility of the learned models (highly non-smooth decision boundaries) rather than the explanation methods. This is a critical point and has to come out more clearly in the paper.